# Critical Flux and Fouling Analysis of PVDF-Mixed Matrix Membranes for Reclamation of Refinery-Produced Wastewater: Effect of Mixed Liquor Suspended Solids Concentration and Aeration

**DOI:** 10.3390/membranes12020161

**Published:** 2022-01-28

**Authors:** Erna Yuliwati, Ahmad Fauzi Ismail, Mohd Hafiz Dzarfan Othman, Mohammad Mahdi A. Shirazi

**Affiliations:** 1Program Study of Chemical Engineering, Faculty of Engineering, Universitas Muhammadiyah Palembang, Jalan A.Yani 13 Ulu Kota, Palembang 30263, Indonesia; 2Advanced Membrane Technology Research Center, Universiti Teknologi Malaysia, Skudai, Johor Bahru 81310, Malaysia; fauzi@petroleum.utm.my (A.F.I.); hafiz@petroleum.utm.my (M.H.D.O.); 3Membrane Industry Development Institute, Tehran 3716137861, Iran; che.shirazi@gmail.com

**Keywords:** critical flux, aeration, mixed matrix membrane, suspended solids removal

## Abstract

Fouling tends to cause a significant increase in hydraulic resistance, decreased permeate flux, or increased transmembrane pressure (*TMP*) when a process is operated under constant *TMP* or constant flux conditions. To control membrane fouling and maintain sustainable operation, the concept of critical flux has been discussed by several researchers. Various fouling mechanisms, such as macromolecule adsorption, pore plugging, or cake build-up, as well as hydrodynamic conditions, for example aeration, can take place at the membrane surface. This study aimed to investigate the effects of mixed liquor suspended solid (MLSS) concentration and air bubble flow rate (ABFR) on the critical flux and fouling behavior, when treating refinery-produced wastewater. To determine the critical flux values, the experimental flux-steps were the following: (1) the filtration began with a 30 min step duration at a low flux (10 to 20 L/m^2^h); (2) at the end of this step (after 30 min), the permeate flux was increased, (3) this step was repeated until the *TMP* did not remain constant at the constant permeate flux, (4) the critical flux was then achieved. A critical flux model with an R^2^ of 0.9 was, therefore, derived, which indicates that the particle properties were regulated by the suspended solids. The increase of MLSS concentration, from 3 mg/L to 4.5 mg/L, resulted in a decrease of the permeate flux by 18%. Moreover, an increase in ABFR, from 1.2 mL/min to 2.4 mL/min, increased the permeate flux, but this decreased with a greater flow rate of aeration. To assess the stability and reversibility of fouling during critical flux (*J_c_*) determination using a mixed matrix membrane, flux-step methods were utilized. A step height of 14.3 L/m^2^h and 30 min duration were arbitrarily chosen. The flux increased to 32.5 L/m^2^h with a slight increase of trans membrane pressure (*TMP*), while the rate of increase became significant at a higher flux of 143.6 L/m^2^h, due to fouling. Overall, this study proved that the response of MLSS concentration and aeration affected the membrane performance, based on the critical flux and fouling behavior.

## 1. Introduction

Membrane applications have been increasingly used in wastewater filtration, due to their promise for handling water scarcity. The properties of feed have also a major impact on membrane fouling because fouling can affect the permeate quality and operating costs. Fouling is also a major limitation to their broader implementation. Particulate matters, inorganic and organic materials (e.g., sulfide, phenols, free-chlorine, and ammonia nitrogen), are potential contributors to membrane fouling in refinery wastewater treatment [1]. A fouling layer is composed of suspended solids; inorganic and organic complexes formed on the membrane surface. The properties of the fouling layer largely control the membrane performance. Preventing or reducing the formation of this fouling layer by using the critical flux concept, or making the layer formation reversible, could enhance the performance of the membrane processes. The reversibility of the fouling layer, as well as the critical flux of suspensions, appear to be dependent on hydrodynamic conditions and physicochemical properties [2,3]. The critical flux can be calculated using the flux-step and pressure-step methods, which indicate the fouling start point. Fouling tends to cause a significant increase in hydraulic resistance, decrease the permeate flux, and enhance the transmembrane pressure (*TMP*), when the process is operated under constant *TMP* or constant flux conditions. Variation of the fouling mechanism, such as macromolecule adsorption, pore plugging, or cake build-up, can take place at the membrane surface [4]. To control membrane fouling and maintain sustainable operation, the concept of critical flux was first introduced in 1995 by Field et al. [5]. The critical flux concept is that, on start-up, there exists a flux below which a decline of flux with time does not occur. Three major approaches can be identified to a specific membrane process, due to the complexities of the fouling phenomena. These are hydrodynamic (changing flow regime across the membrane surface), surface modification (changing the surface or foulant affinity), and regular cleaning.

The pH change, ionic strength, and suspended solids concentration, as well as the process conditions of submerged ultrafiltration (e.g., air bubbles flow rate, temperature, pressure), affect the critical flux and the properties of this fouling layer. The effect of continuous aeration on the fouling layer in submerged membrane ultrafiltration has been discussed by several researchers [6,7]. However, the presented information about these effects has been unclear. The critical fluxes of various feed materials have been studied by Liang et al. (2020) [8]. They found that the critical flux of milk protein suspensions, sodium caseinate, and whey protein depended on the hydrodynamics and their physicochemical properties. Wu et al. (2021) studied critical flux determination using the flux-step method in a submerged membrane bioreactor for synthetic sewage wastewater [9]. They found that the specific hydraulic parameters calculated from the flux-step could be nearly identical to the critical flux values. This value indicated the start point of fouling but did not yield absolute predictive permeability data for extended operation.

In the membrane process, Zheng et al. (2015) studied the crossflow velocity induced by coarse and fine bubbles [10]. The bubbles were generated using 2.0 mm and 5.0 mm diffuser sizes, respectively. They noted that the fine bubbles produced appeared to initiate higher cross-flow velocities and were distributed uniformly at the same aeration intensity. These conditions provided better fouling control, to prolong the membrane operation. The importance of module design for process efficiency in aerated submerged membrane processes was reported by researchers, whereby an appropriate membrane module could maximize the hydrodynamic effects induced by bubbles [11,12,13]. Here, we studied the effects of mixed liquor suspended solid concentration and air bubble flow rate on the critical flux and fouling behavior of refinery-produced wastewater.

## 2. Experimental

This experimental work consists of two investigations of critical flux and fouling mechanisms, using submerged ultrafiltration. The synthetic and real refinery-produced wastewater were also used to measure COD, TSS, and NH_3_-N removal.

### 2.1. Preparation of Ultrafiltration System

The ultrafiltration system used in this study consisted of a reservoir of 14 L volume, hollow fiber modules, a peristaltic pump, and an aerator system, as shown in Figure 1. The system was fed with in-house synthetized refinery-produced wastewater, the main properties of which are listed in Table 1. The modified PVDF hollow fibers were prepared by employing PVDF (Kynar^®^740 Arkema Inc., Philadelphia, PA, USA) of 16 wt.% in DMAc (Aldrich, Chemical Synthesis grade, Merck, Darmstadt, Germany, 99%) at different organic additive titanium dioxide (TiO_2_) (Merck, Germany) concentrations of 10 wt.%, purchased from Sigma-Aldrich, Merck, Germany, and Lithium chloride (LiCl) (Merck, Germany) at 3 wt.% (purchased from Merck, Germany) of the weight of PVDF, respectively. The dope solution was pressurized through a spinneret with a controlled extrusion rate, while the internal coagulant was adjusted to 1.4 mL/min, and the airgap length was set at 2 cm. The fibers were exuded from the tip of the spinneret and were then guided through the two water baths, to release the extra solvent, for three days. The fibers were then post-treated using glycerol at 10 wt.% for 24 h. After drying, the fibers were ready for making a module. The fibers, that had a total effective area of approximately 11.23 dm^2^, were then immersed in the membrane reservoir and transmembrane pressure was controlled on the permeate side. To enhance the membrane hydrophilicity of PVDF membranes, LiCl and TiO_2_ were added to the spinning dope during the membrane preparation process, to improve membrane water filtration. The structure of the porous membrane and TiO_2_ nanoparticle hydrophilicity were directly correlated with membrane porosity, which was responsible for the higher liquid uptake. Details of the membrane fabrication process and the property determination procedure can be found in a previous study [14]. The selected main properties of the used wastewater in this study have been analyzed as shown in Table 1.

The experiments were carried out in a vacuum condition, which was produced when permeate was withdrawn from the open end of the fibers using a peristaltic pump (Master flex model 7553-79, Cole Palmer, Illinois, USA). The volume of feed in the membrane reservoir was maintained throughout the experiment. The generation of air scouring bubbles was advantageous for exerting shear stress, to minimize the deposition of particles on the membrane surface during the filtration process. The air bubbles provided direct shear, inducing a secondary flow of liquid, and moving the membrane. An increase in air bubbles’ flow rate and crossflow velocity suppressed fouling and enhanced the permeate flux. The flow pattern of air bubbles within the submerged ultrafiltration is illustrated in Figure 1. However, Tana et al. (2014) observed an optimum aeration rate beyond which a further increase had an insignificant effect on fouling suppression [15].

The permeate water volume was collected using a graduated cylinder. Periodic cleaning was conducted after finishing each batch of filtration. Then, 1000 mg/L NaOH aqueous solution was used as the alkaline cleaning agent for washing the membranes in the first 20 min of each cleaning step, followed by 5 min of rinsing with pure water. After completing filtration, the membranes were cleaned with a soft sponge to remove any accumulated particles that might be embedded on the membrane surface.

The experimental conditions were set as a constant hydraulic retention time (300 min) and room temperature, but with different MLSS concentrations and air bubble flow rates, as listed in Table 2. Meanwhile, the membrane properties and operating characteristics of the submerged ultrafiltration are described in Table 3.

### 2.2. Membrane Characterization

A field emission scanning electron microscope (FESEM) (Hitachi High Technology Co., Ltd., Tokyo, Japan) was used to examine the morphology of the PVDF hollow fiber membrane prepared. Before analysis, the membrane samples were first immersed in liquid nitrogen and carefully fractured. The samples were then coated with sputtering platinum before testing. FESEM photomicrographs of the cross-section and the outer surface of the hollow fiber membranes were taken at various magnifications. Moreover, atomic force microscopy (AFM) (SPA-300 HV, Park System Inc., Santa Clara, CA, USA) was used to illustrate the surface roughness of the clean and fouled membranes. Asymmetric porous membranes were characterized using a determination of porosity and average pore radius [16].

### 2.3. Critical Flux Evaluation

The basis for the membrane filtration model in the presence of fouling is given by Equations (1)–(8). The driving force might be reduced by osmotic pressure, due to concentration polarization, whilst the hydraulic resistance may be greater than that of the membrane itself, due to material accumulation on the membrane surface and/or in the membrane pores. These phenomena affect the relationship between flux and transmembrane pressure (*TMP*). Moreover, the membrane resistances can be considered in series, as follows.
(1)$$J=\frac{\Delta P}{\mu \;\left(R_{m}+R_{cp}+R_{c}+R_{ad}\right)}$$
where *J* is the permeate flux (L/m^2^h), Δ*P* is the transmembrane pressure (Pa), *µ* is the permeate viscosity (Pa s). *R_m_* is the intrinsic membrane resistance, *R_c_* is the cake resistance, *R_cp_* and *R_ad_* are the resistance due to permanent fouling layer and adsorption with units in m^−1^, respectively.

To determine the total resistance
(2)$$R_{t}=R_{m}+R_{cp}+R_{c}+R_{ad}$$
(3)$$R_{f}=R_{cp}+R_{c}+R_{ad}$$
thus,
(4)$$R_{t}=R_{m}+R_{f}$$

Under vacuum pressure and a feed temperature of 25 °C, the pure water flux (PWF) was calculated in the collected permeate water within 45 min. The distilled water was replaced with synthetic refinery wastewater or real synthetic wastewater at the same operating conditions. Permeate was periodically taken for each 30 min throughout the 300 min filtration time. After 300 min of filtration, the permeate flux was recorded and labeled as *J_cp_*. The membrane was then rinsed with deionized water, to remove the concentration polarization (CP) layer, for 45 min and the PWF was measured again (*J_c_*). Finally, the membrane was washed with 0.1 M NaOH solution for 20 min and the PWF was reevaluated (*J_ad_*).

As the resistance of *R_c_* and *R_ad_* can be generally removed using deionized water flushing and chemical cleaning, respectively, individual resistances due to different mechanisms can be calculated using the following equations:(5)$$R_{t}=\frac{\Delta P}{\mu }$$
(6)$$R_{cp}=\left(\frac{\Delta P}{\mu J_{ad}}\right)-R_{m}$$
(7)$$R_{ad}=\left(\frac{\Delta P}{\mu J_{c}}\right)-R_{m}-R_{cp}$$
(8)$$R_{c}=\left(\frac{\Delta P}{\mu J_{cp}}\right)-R_{m}-R_{cp}-R_{ad}$$
where *J_w_* is the initial water flux (m/s), *J_c_* is the water flux after removing the cake layer by flushing with deionized water and chemical cleaning with a 0.1 M NaOH (m/s), and *J_cp_* is the flux of refinery wastewater at the end of filtration (m/s).

According to Darcy’s law, the flux of UF/MF membrane filtration can be expressed as follows [17]:(9)$$J=\frac{V}{A\;dt}=\frac{TMP}{\mu \left(R_{T}\right)}$$
where *J* is the permeate flux (L/m^2^h), *TMP* is transmembrane pressure (Pa), *μ* is the viscosity (Pa s), and *R_T_* is the total membrane resistance.

The critical flux was determined according to the flux step method, consisting in setting increasing values of flux and registering the relative transmembrane pressure (*TMP*) variations, as shown in Figure 2 [18]. By considering the fouling mechanisms, a strong form of critical flux was developed to discriminate no-fouling conditions (whereas *R_m_* is only the resistance in Equation (1)) from fouling conditions, where other resistances (*R_c_*) also apply. The form of critical flux can define the application of resistances [19].

To calculate the critical flux values, the experimental flux-steps were the following: (1) the filtration begins with a 30 min step duration at a low flux (10 to 20 L/m^2^h); (2) at the end of this step (after 30 min), the permeate flux is increased; (3) this step is repeated until the *TMP* does not remain constant at a constant permeate flux; (4) critical flux is then achieved. Two *TMP* values were considered for each step, namely the initial *TMP*, which corresponds to the initial sudden increase of filtration resistance, and the final *TMP*, which is the *TMP* at the end of the step. From these two *TMP* values, two parameters connected to fouling can be evaluated. These are the average *TMP* and the rate of *TMP* increase (dTMP/dt). The critical flux was assumed to be the flux at dTMP/DT ≥ 0.5 [16,19].

In particular, the concept of critical flux can also be illustrated as a function of aeration. A relationship is observed between the critical flux and the air bubble flow rate, as shown in Figure 3. The critical flux of the system corresponds to an equilibrium state, between the drag torque force of the particles on the membrane and the forces driving the particles off the membrane. This was shown in the interaction of suspended solid particles and air bubbles flow rate; as the critical flux characterization depends on the experimental conditions [20].

### 2.4. Fouling Mechanism

The filtration theory was described, in particular, by Poiseuille’s law, which could distinguish many models, such as blocking models and cake filtration theory. The blocking model was first put forward by Hermans and Breede (1935) and Hermia (1982) who explained the membrane fouling method [21]. The membrane fouling mechanism was analyzed by applying a mathematical model to Hermia’s model. Hermia’s model provided a comprehensive fouling prediction model, well-equipped with four different fouling mechanisms [22]. The general law was used to explain the blocking models, namely, complete blocking, standard blocking, intermediate blocking, and cake formation. The mechanism-illustration model is shown in Equation (10).
(10)$$\frac{d^{2}t}{dV^{2}}=k{\left[\frac{dt}{dV}\right]}^{n}$$
where *dt* is the filtration time (s), *dV* is the filtered volume (m^3^), *k* is the hydraulic resistance coefficient (is a constant), *n* is a blocking index, which can take different values according to the fouling mechanism involved. *n* equals 0, 1, 3/2, and 2 to define, respectively, cake, intermediate, standard, and complete blocking filtration.

As analyses of membrane filtration are normally performed in terms of flux, it should be noted that Equation (10) can be presented in an alternative form $\frac{dV}{dt}$
*= A J*, where *A* is the total membrane active area and *J* is the permeate flux, which is illustrated in Equation (11):(11)$$\;\frac{1}{A^{2}J^{3}}\frac{dJ}{dt}=k{\left[\frac{1}{A\;J}\right]}^{n}$$

For the cake model (*n* = 2 for complete blocking filtration), the analysis of equilibrium distribution contact (*k*) can be expressed as shown in Equation (12) [23]:(12)$$k=C_{F}\frac{\rho_{c}\eta }{\Delta P}$$
where $\rho_{c}$ is the cake mass per unit of permeate volume, *η* is the feed solution viscosity, and *C_F_* is the apparent specific resistance of the cake. To calculate *C_F_*, it has been assumed that the deposited mass per unit of filtered volume, *ρ_c_* is equal to the feed oil/water concentration. This is strictly true only for dilute concentrations [24]. The form of complete blocking filtration can be expressed as follows:*lnJ* = *lnJ*_0_ − *K_c_ t*(13)
where *K_c_* is the constant of each type of blocking mechanism model and *J_v_*_,0_ is assumed to be the flux at an initial time.

Membrane fouling is a complex phenomenon, where the permeate flux declines drastically due to chemical and physical factors [24]. The flux drop due to membrane fouling (intermediate blocking) can be summarized as follows [25]:(14)$$J_{v}=\frac{1}{A}=J_{v,0}{\left(1+K_{t}\right)}^{n}$$
where *J_v_* is the permeate flux (L/m^2^h), *A* is the effective area of membrane (mm^2^), *V* is the permeate total volume (L), *t* is the time (h), and *J_v_*_,0_ is the initial flux of the resulting submerged membrane after the blocking of the initial pores [26].

Since the fouling appears on the surface and/or an internal pores, the fouling mechanisms are summarized by two models, namely, the standard blocking model (also called pore constriction) and the pore-blocking model, as shown in Figure 4.

The standard blocking model takes place when fouling occurs along the pore walls, minimizing pore diameter, while the pore density remains constant. The blocking model occurs when the number of pores that become plugged is enhanced proportionally to the filtrate volume, whereas the pore diameter remains constant [26].

### 2.5. Water Analysis Methods

To study the effect of MLSS concentration on submerged membrane ultrafiltration, fouling is not as obvious as the air bubbles flow rate effects, mainly due to the complexity and variability of the biomass components. While the extra polymer substances and other biomass characteristics are not accounted for, the increase in MLSS concentration alone has a mostly negative effect on the flux obtained in a SS MBR [27], the stabilized permeation rate [28], and on the limiting flux [29]. In addition, the flux was increased when changing the MLSS concentration from 3.5 to 10 g/L, with an associated change in mean floc size from 200 to 50 μm [29]. Although the same type of membrane was used in both studies and the hydraulic conditions were similar, the flux values reported by Huang et al. (2014) differed significantly, namely 221 L/m^2^h for MLSS of 4 g/L and 321 L/m^2^h for MLSS of 2.5 g/L [30]. This observation demonstrates the importance of carrying out tests under the same conditions when assessing hydraulic/hydrodynamic parameters and impacts.

Meanwhile, the air bubble flow rate (ABFR) and cross-flow velocity (CFV) influence fouling and increase flux. Although most of the studies on *J_c_* were based on sidestream (SS) operation, some studies were carried out with a submerged membrane process or with ideal feed solutions. Zhinun et al. (2020) suggested that an increase in airflow rate at the membrane surface limits fouling [28]. However, Schwarze (2017) observed an optimum aeration rate, beyond which a further increase did not affect fouling suppression [31]. Details of the phenomena occurring during air sparging have been extensively reported. It was described as the use of bubbles flowing on the outside of the fiber. The mechanism of fouling control was also discussed in detail and identified as follows: (1) bubbles induced secondary flow; (2) physical displacement of the concentration polarization layer; (3) pressure pulsing caused by passing bubbles; (4) increase in superficial cross-flow velocity [31].

The parameters of Chemical Oxygen Demand (COD), Total Suspended Solids (TSS), and ammonia nitrogen (NH_3_-N) concentrations were measured using a spectrophotometer (DR 5000, HACH) using the standard procedures. During the operation with high organic loading rates, the permeate was evaluated daily, and sampling was carried out three times a week. The values of COD, TSS, and NH_3_-N were calculated with Equations (15)–(17), while COD removals were measured using gravimetric method EPA-1664 and calculated using the following equation:(15)$$\mathrm{COD}\;\mathrm{removal}\;\left(\%\right)=\frac{{\mathrm{COD}}_{0}-\mathrm{COD}}{{\mathrm{COD}}_{0}}$$
where COD_0_ and COD are the initial concentration of synthetic refinery wastewater and the concentration of permeate produced.
(16)$$\mathrm{TSS}\;\mathrm{removal}\;\left(\%\right)=\frac{{\mathrm{TSS}}_{0}-\mathrm{TSS}}{{\mathrm{TSS}}_{0}}$$
where TSS_0_ and TSS are the initial concentration of synthetic refinery wastewater and the concentration of permeate produced.
(17)$$\mathrm{NH}3-N\;\left(\%\right)=\frac{\mathrm{NH}3-N_{0}-\mathrm{NH}3-N}{\mathrm{NH}3-N_{0}}$$
where $\mathrm{NH}3-N_{0}\;$. and NH3−N are the initial concentration of synthetic refinery wastewater and the concentration of permeate produced.

## 3. Results and Discussions

### 3.1. Microscopic Analysis Using FESEM and AFM

Figure 5 shows the FESEM photomicrographs of the clean PVDF hollow fiber membranes. The morphology of the membrane was observed with the addition of TiO_2_ nanoparticles. TiO_2_ nanoparticles have a high specific area and hydrophilicity, which influences the mass transfer during the spinning process. The cross-sectional images consist of finger-like macro voids extending from both the inner and outer wall of the hollow fiber, and an intermediate sponge-like layer. Yuliwati and Ismail (2011) found an average outer diameter ranging from 1.1 to 1.08 mm and an inner diameter from 0.55 to 0.57 mm [31]. The average pore size was 34.05 nm, with a porosity of 88.42% [22].

Assuming that the residual fouling corresponds to irreversible fouling, it appears that reversible fouling was responsible for 15.35% and 25.17% of the total flux loss caused by total fouling for MLSS of 3 and 4 g/L, respectively, at an air bubble flow rate of 2.4 mL/min. This indicates that irreversible fouling represents the main contribution to the fouling mechanism. In order to observe the surface morphology of the membrane with irreversible fouling, FESEM analyses were also conducted, to illustrate the fouled surface membrane.

FESEM images of the fouled membranes are shown in Figure 5 and Figure 6. In these figures, most large pores are not visible anymore, due to the presence of particles of different shapes and sizes combined with a polymer matrix. Figure 5a,b shows the layer is porous, contrary to that observed for Figure 6a,b. A thicker and denser surface layer, consisting of many particles of suspended solids and aggregates, is observed. This is caused by a progressive penetration into the pores of small particles of equivalent size (cell fragments) and some dissolved macromolecules.

Meanwhile, the membrane surface 3-D images were also observed using AFM. The high peaks seen as bright regions characterize the nodules, while the pores are seen as dark depressions, as shown in Figure 7 and Figure 8. Figure 7 shows the AFM images expressed nodule-like structures of the outer surfaces of clean membranes. After filtration, the outer surface of both membranes seemed smoother, which was caused by filtration caking. The feed solution that contained mixed liquor suspended solids concentration of 4.5 g/L had larger grains than those at 3 g/L, due to the foulants deposited on the surface membrane (Figure 8).

The size of nodule aggregates increased on both fouled surfaces, as shown in Figure 8. The results indicate that the higher MLSS concentration promoted more significant irreversible fouling, faster reversible cake establishment, and consequently decreased the permeate flux during filtration. This agrees with the study by Lukina et al. (2016), which showed suspended solids participated in the membrane fouling, which caused deposition, pore blocking, and irreversible fouling [24]. This promotes the formation of a filtration cake at the beginning of filtration, due to the reduction of pore density and pore diameter of the outer surface of the membrane.

### 3.2. Membrane Fouling Mechanism

In order to assess the stability and reversibility of fouling during critical flux (*J_c_*) determination, using a submerged membrane ultrafiltration fed with synthetic refinery wastewater, flux-step methods were utilized. A step height of 14.3 L/m^2^h and a 30 min duration were arbitrarily chosen, as shown in Figure 9. With this, the flux increased to 32.5 L/m^2^h at a slight increase of *TMP*. Meanwhile, the increase of *TMP* became significant at a higher flux of 143.6 L/m^2^h, due to fouling. During this flux step method, *TMP* values were shown that corresponded to values of descending phase more than ascending phase. For example, at the initial flux-step of 32.5 L/m^2^h, TMP(P)_ave_ was 80 and 30 mbar (absolute) for the ascending and descending phases, respectively. When consecutive cycles were carried out, without intermediate cleaning (i.e., neither backwash nor chemical cleaning), the critical hydraulic performance appeared to change little following the first cycle, with *TMP* values remaining at 0.5 bar (absolute) for the lowest flux value of 14.3 L/m^2^h for both the second consecutive cycles. These observations indicated the formation of an initial irreversible fouling layer after the first flux-step cycle, on which some reversible fouling forms.

The behavior of the three parameters was investigated for the membranes, in terms of the initial water permeability, during step method and after membrane cleaning with MLSS of 3 g/L, as shown in Figure 10. This figure exhibits a comparable water permeability. The clean water permeability was slightly higher than that after membrane cleaning. It was indicated that most probably, the deposition of materials on the membrane surface and/or in the membrane pores could be removed almost completely with the cleaning procedure. The PVDF hollowfiber membrane was exposed to the suspended solid particles with a skin layer that had a small pore size, if compared to the dimension of those particles. Consequently the PVDF hollowfiber membrane was less influenced by the deposition and adsorption phenomena of difficult to remove materials from its porous structure.

The first part of the filtration experiments met the standard blocking laws and the following part, up to the end of the run, met the cake filtration model. One of the results is shown in Figure 11. The standard blocking period was very short, and the cake filtration predominated in all filtrations. Moreover, all experimental data could be described by the cake filtration model.

### 3.3. Critical Flux, Resistance, and Filtrate Analysis of Refinery-Produced Wastewater

Critical flux in the ultrafiltration membranes was obtained using the flux stepping method. In this study, the flux stepping method was used to determine the critical flux. The filtration experiment was conducted with modified PVDF membranes for the synthetic refinery wastewater [22]. The fluxes are shown in Table 4 quantitatively, and the results are shown in Figure 12.

Table 4 shows the critical flux test conducted with MLSS of 3 g/L and ABFR of 2.4 mL/min.

As shown in Figure 12a, the flux decreased significantly in the first 30 min of filtration, then linearly with the increase of hydraulic retention time (HRT) until the HRT reached 210 min. It changed non-linearly with a further increase of HRT. This can be further illustrated by Figure 12b, where the total fouling resistance (*R_f_*) increases with the increase of HRT in all the process conditions for both membrane samples.

The operating flux was less than the critical flux, i.e., the HRT is in the range of 0–180 min as shown in the A zone. The rising tendency of the total fouling resistance with the increase of HRT was inconspicuous. In contrast, as illustrated in the B zone of Figure 12b while the operating flux was close to the critical flux point, the concentration polarization phenomenon still existed. Once the flux exceeded the critical point, there was concentration polarization and coexistent membrane fouling, which the *R_f_* started to increase significantly. This condition forces the large molecules or suspended solid particles to interact with the membrane surface and cause particle adsorption and aggregation on the membrane surface or in the membrane pores. Therefore, the membrane pores gradually diminish, and this produces membrane fouling. Since the operating flux is more than the critical flux, i.e., the HRT is in the range 210–270 min (zone C), the increase of *R_f_* starts to accelerate. Since the particles were adsorbed and aggregated ceaselessly in the membrane pores, the fouling of the membrane increased significantly. The formation of the cake layer and the irreversible membrane fouling became obvious.

To understand the phenomenon of critical flux and the involvement of the membrane fouling in the operating process in depth, the different resistances in the membrane filtration process were analyzed, and the results are described in Figure 12. Moreover, the increase of (*R_t_* − *R_m_*)/*R_t_* values with increasing HRT is illustrated in Figure 13. In A zone, the value of (*R_t_* − *R_m_*)/*R_t_* was low and unchanged. This implies that membrane fouling did not occur and only the effect of concentration polarization occurred. In the B zone, the critical flux point was achieved, and then membrane fouling appeared and became serious, with a further increase of operating flux.

The value of (*R_t_* − *R_m_*)/*R_t_* varied obviously, from 30.21% to 42.90%. Moreover, the value of (*R_t_* − *R_m_*)/*R_t_* varied gradually, from 43.80% to 44.60% in the HRT range of 210 to 300 min, as shown in the C zone. At the same time, the values of *R_t_*/*R_m_* increased with the increasing of the operating flux, from 1.50 to 1.75. The value of *R_t_*/*R_m_* being equal to 1 means that the total resistance was the intrinsic resistance of the membrane; thus, there was no membrane fouling resistance. However, the operating flux exceeded the critical flux point and became larger, and the value of *R_t_*/*R_m_* increased from 1.79 to 1.85. This means that the membrane fouling resistance gradually increased and became the dominant resistance. Therefore, the intrinsic resistance of the membrane can be ignored in the total membrane resistance.

### 3.4. Effect of Air Bubbles Flow Rate on the Performance of Membrane Ultrafiltration

An increase of membrane fouling with increasing MLSS concentration was found in many literature works by various researchers, but some other studies revealed no effect of MLSS concentration on fouling up to a threshold concentration [32]. The effect of air bubbling used in submerged membrane systems was investigated. The continuous airflow rate enhanced the membrane critical flux and, thus, minimized the fouling on the surface membrane [32]. It is known that membrane fouling can be considered from a critical flux point of view. Figure 14 shows the trend in *dP*/*dt* for various air bubble flow rates. The results indicated that the use of an air bubble flow rate of 2.4 mL/min caused an increase of flux more than that of air bubble flow rates of 1.2 and 3.0 mL/min. A degree of suppression of irreversible fouling occurred at air bubble flow rates of 2.4 mL/min, due to the achieved highest flux. Moreover, this meant that the aeration can be tuned according to the permeate flux, to reduce the power consumption related to the air scouring.

### 3.5. Effect of MLSS Concentration on the Critical Flux and Fouling Analysis

The fouling rate under different specific MLSS concentrations in the feed wastewater with air bubbles flow rate of 2.4 mL/min is shown in Figure 15. It was observed that during experiments the flux for feed solution with MLSS concentration of 3 g/L became higher than that of 4.5 g/L. This fact suggests a higher tendency of the suspended solid concentration to interact with the membrane surface and the ability of air bubbling to enhance the permeate flux. The critical flux values on submerged ultrafiltration became lower by 18%. The *R_f_* value of this test was lower; thus, leading to a remarkable decrease of the total resistance, especially at low MLSS concentration. For both MLSS concentrations, the fouling tended to be similar trend.

To assess the stability and reversibility of fouling, flux-steps were conducted in both ascending and descending phases, from 4.5 L/m^2^h to 144.5 L/m^2^h, for a flux-step duration of 15 min. These experiments were carried out at the MLSS concentrations of 3 and 4.5 g/L and at a constant air bubble flow rate of 2.4 mL/min. The *P_ave_* values obtained during the descending phase of the cycle were greater than those of the corresponding ones observed during the ascending phase. In this experiment, it was found that the use of an MLSS concentration of 3 g/L in the feed solution demonstrated a higher permeability than 4 g/L. The flux first increased proportionally with increasing *P_ave_*, and tended to level off more quickly at higher MLSS concentrations during the ascending phase, due to fouling. The effect of fouling on the membrane flux was also apparent by observing the behavior of the membrane flux as a function of the applied *P_ave_* during the descending phase. Moreover, a hysteresis curve for the membrane tested was also observed using process conditions with different air bubble flow rates, as shown in Figure 16.

Figure 17 showed the trend in *dP*/*dt* for a constant air bubble flow rate for two types of refinery wastewater, namely, synthetic, and real refinery-produced wastewater. Significant variations were observed, in terms of membrane flux recovery, as expressed by the recovery factor *dP*/*dt*. The data showed an insignificant change of the first *dP*/*dt* values, although non-zero, up to certain flux value, which was 0.3 and 0.4 mbar.min^−1^ for synthetic and real refinery produced waste water, respectively.

For higher fluxes, an exponential relationship between *dP*/*dt* and the flux could be discerned and the comparable fouling rates, i.e., *dP*/*dt* values at the same flux, for real refinery-produced wastewater were observed to be much lower than those measured for the synthetic wastewater. Although, both wastewaters tended to have similar trends of flux as a function of *dP*/*dt*. This can be more explained with the data in Table 5. The removal of the main parameters of permeate was calculated, and similar results were achieved.

## 4. Conclusions

A PVDF-hollowfiber membrane was used to elucidate the critical flux and fouling mechanism. The filtration results met the standard blocking and cake filtration models. The critical flux was observed using the flux-step method in submerged ultrafiltration. The total fouling resistance increased with the increase of time. When the operating flux was below the critical flux, there was only a concentration polarization phenomenon. The membrane resistance increased when increasing the flux. The intrinsic membrane resistance was the dominant resistance, and the filtration process achieved the promised values. However, once the operating flux exceeded the critical flux, the membrane fouled and the rate of resistance accelerated. The fouling resistance was the dominant resistance; thus, the filtration proceeds were smaller.

The critical flux was also significantly affected by the air bubble flow rate, suggesting that an enhanced air bubble flow rate can be used to minimize fouling during hydraulic loads and also enhance the critical flux. FESEM and AFM images indicated that the membrane was covered with foulant on the outer surface of the membrane.

The effect of MLSS concentration and air bubble flow rate was clearly demonstrated. The increase of MLSS concentration resulted in a lower permeate flux. Moreover, an increase in air bubble flow rate increased the permeate flux but decreased with a further increase of air bubble flow rate. Similar results were also described with synthetic and real refinery-produced wastewater as feed solutions. Finally, it is apparent that membrane fouling in submerged membrane ultrafiltration takes place, even at low flux rates, but changes dramatically when the critical flux is achieved. The critical flux value calculated by the flux-step method, thus, indicates the point at which fouling starts to become severe, but does not yield predictive absolute permeability data for extended operations. A closer examination of the *TMP* behavior is required to define the critical flux value more precisely. The synthetic refinery-produced wastewater used in this study generated a mixed liquor of substantially greater fouling propensity than the real refinery-produced wastewater studied.

## Figures and Tables

**Figure 1 membranes-12-00161-f001:**
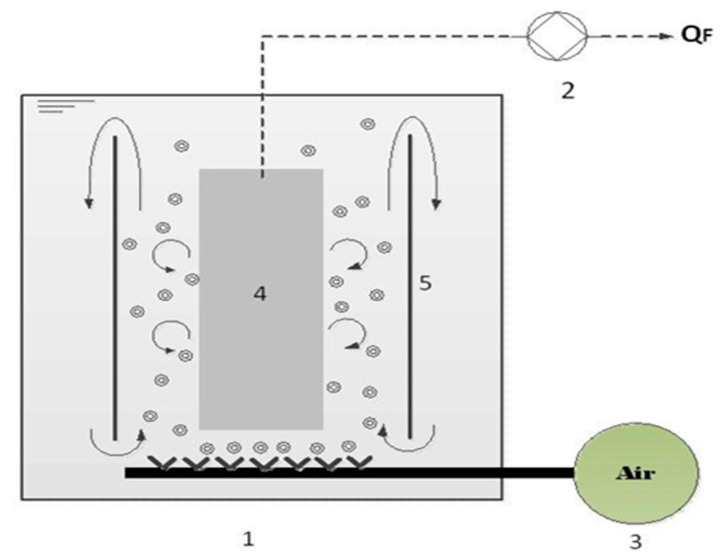
Schematic representation of the air bubble up flow stream in a submerged hollow fiber UF system: membrane reservoir (1); peristaltic pump (2); aerator (3); membrane bundles (4); and partitioned glass (5).

**Figure 2 membranes-12-00161-f002:**
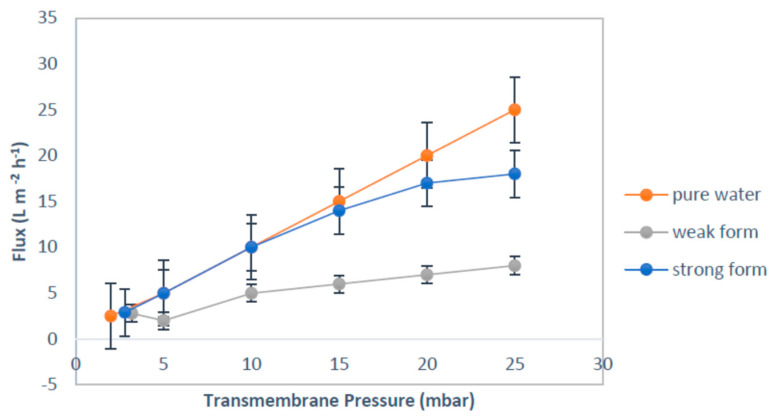
Forms of critical flux.

**Figure 3 membranes-12-00161-f003:**
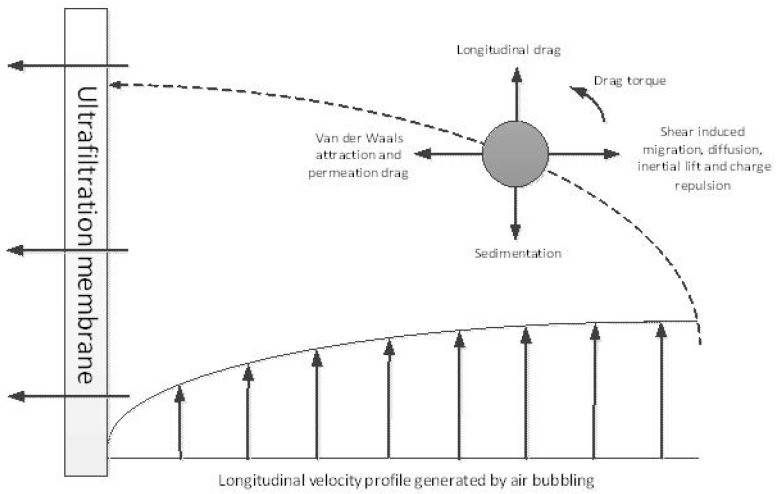
Force equilibrium on a particle or colloid.

**Figure 4 membranes-12-00161-f004:**
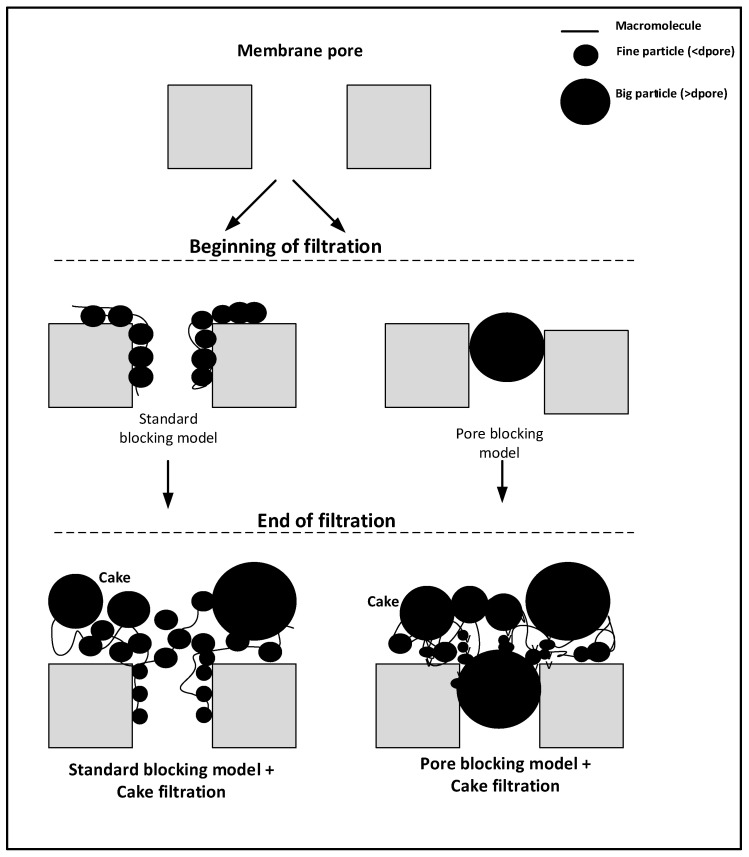
Illustration of standard and pore-blocking models, and cake filtration during membrane filtration.

**Figure 5 membranes-12-00161-f005:**
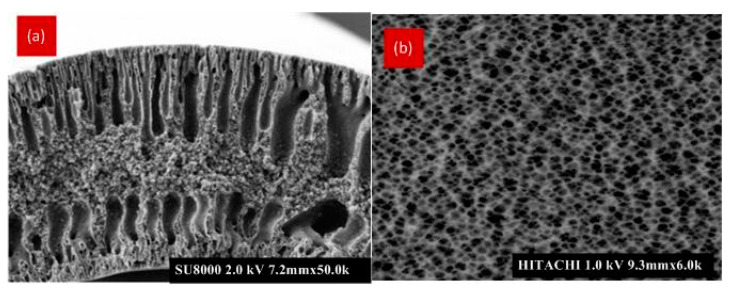
FESEM images of the (**a**) cross section (Mag. 500) and (**b**) outer surface (Mag. 40.0 k) of cleaned membranes.

**Figure 6 membranes-12-00161-f006:**
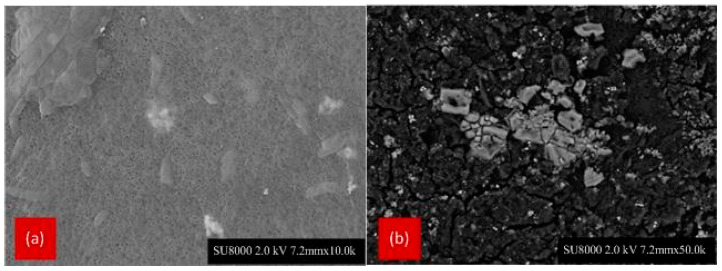
FESEM images of the outer surface of fouled PVDF membranes (Mag. 500) at the refinery wastewater with mixed liquor suspended solids concentrations of (**a**) 3 g/L and (**b**) 4.5 g/L.

**Figure 7 membranes-12-00161-f007:**
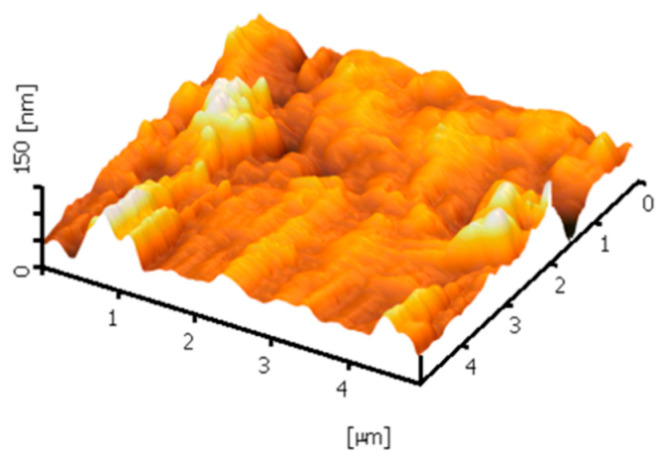
AFM images of the clean outer surface membrane.

**Figure 8 membranes-12-00161-f008:**
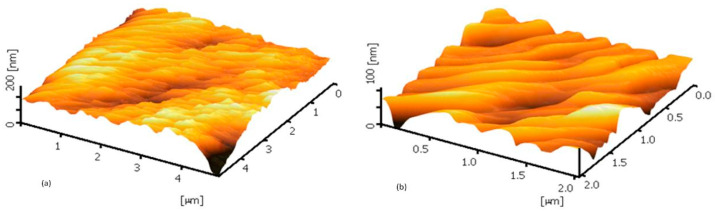
AFM images of the fouled outer surface membranes with MLSS concentrations of (**a**) 3 g/L and (**b**) 4.5 g/L.

**Figure 9 membranes-12-00161-f009:**
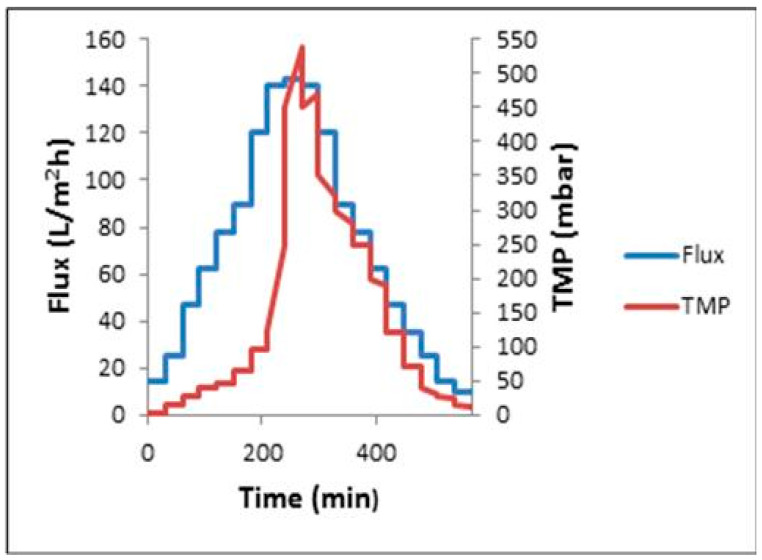
Flux-step method for critical flux determination of synthetic refinery wastewater (MLSS and air bubbles at a flow rate of 3 g/L and 2.4 mL/min, respectively).

**Figure 10 membranes-12-00161-f010:**
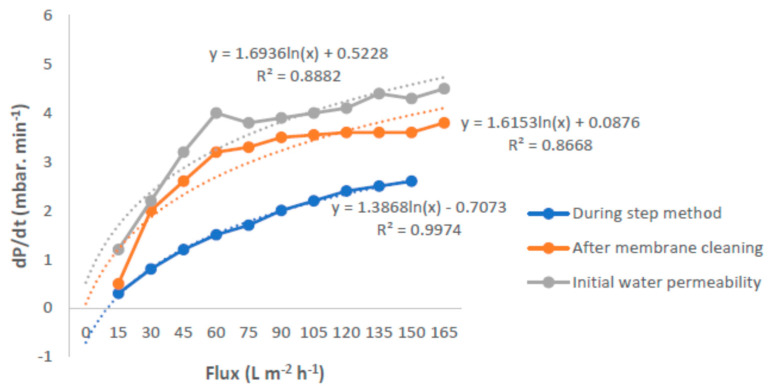
Effect of dp/dt on flux for synthetic refinery wastewater.

**Figure 11 membranes-12-00161-f011:**
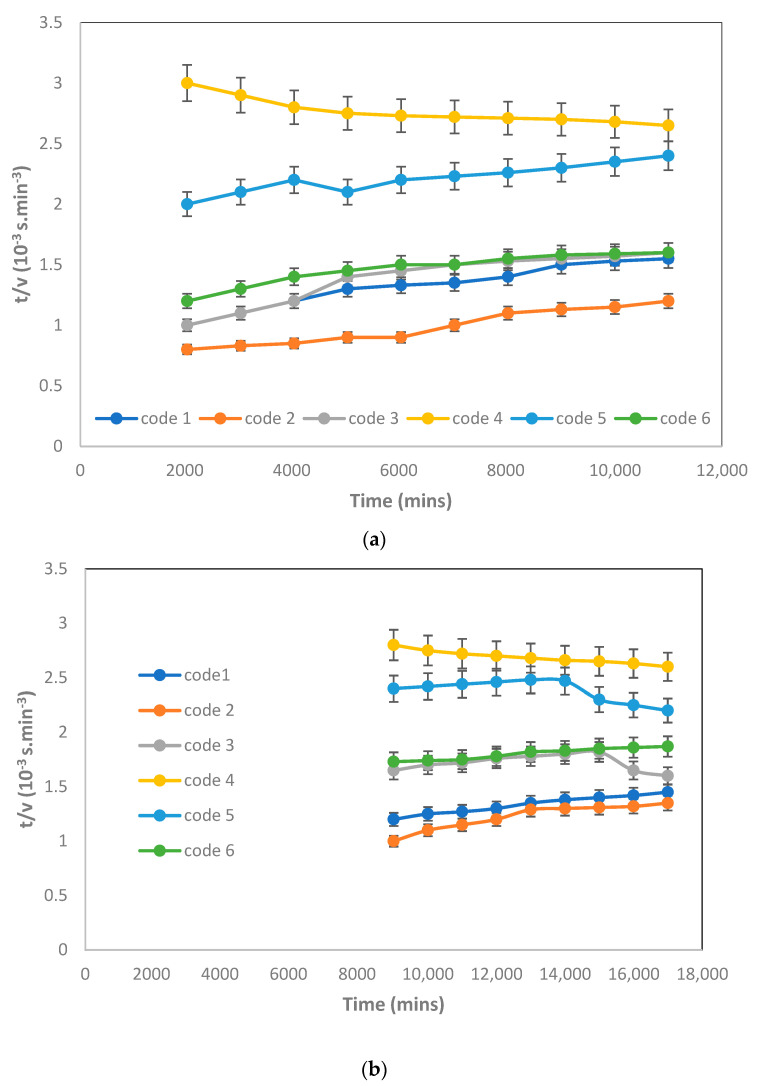
Standard blocking (**a**) (Filtration time was from 0 to 10,800 s) and cake filtration (**b**) (Filtration time was from 10,800 to 18,000 s) at −15 in Hg and room temperature.

**Figure 12 membranes-12-00161-f012:**
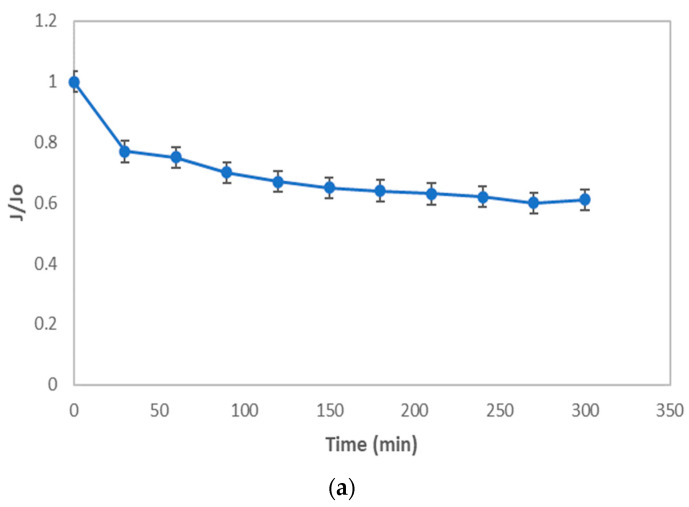
The flux vs. time (**a**) and the fouling rate vs. time (**b**) at room temperature.

**Figure 13 membranes-12-00161-f013:**
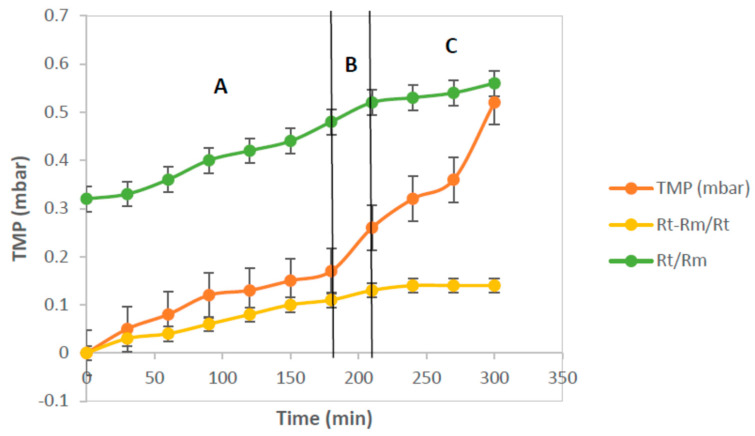
*R_t_* − *R_m_*/*R_t_* and *R_t_*/*R_m_* from process condition (code 3) as a function of the filtration time.

**Figure 14 membranes-12-00161-f014:**
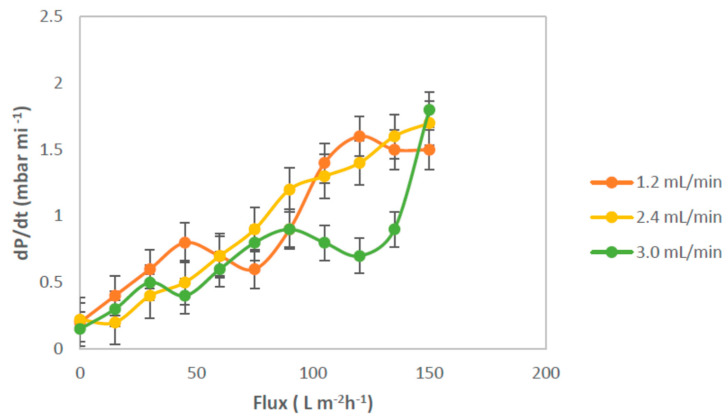
Effect of air bubble flows rate on the permeate flux at an MLSS concentration of 3 g/L.

**Figure 15 membranes-12-00161-f015:**
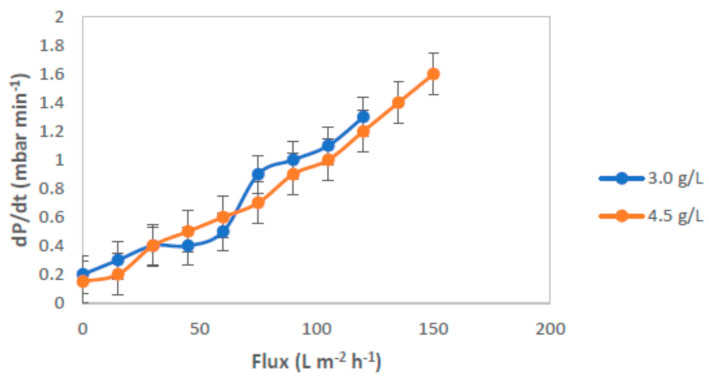
Trend of *dP*/*dt* as a function of flux for MLSS concentrations of 3.0 and 4.5 g/L.

**Figure 16 membranes-12-00161-f016:**
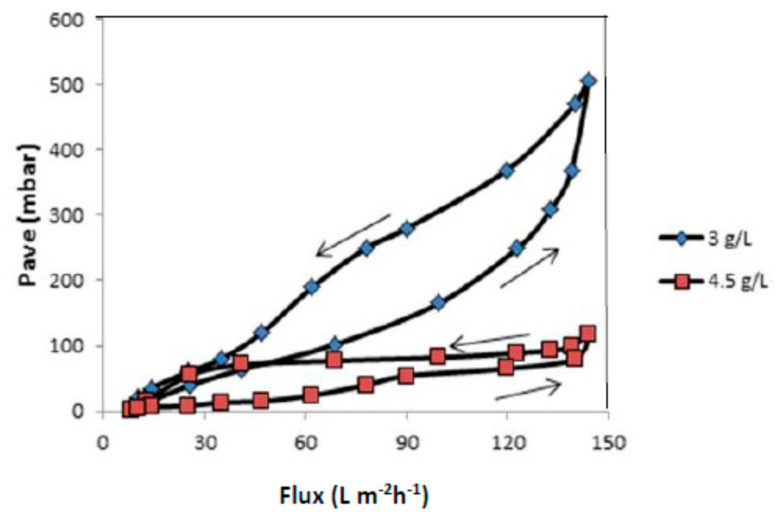
Hysteresis curves for the membrane tested with MLSS concentrations of 3 and 4.5 g/L at a constant air bubbles flow rate of 2.4 mL/min.

**Figure 17 membranes-12-00161-f017:**
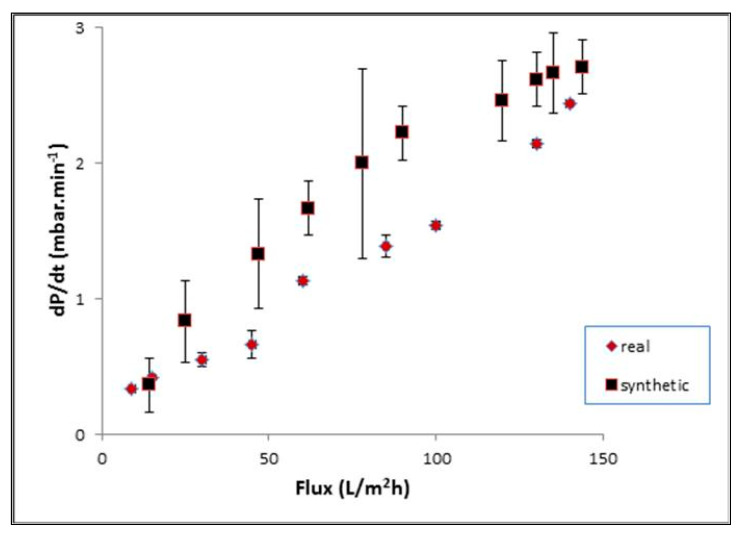
Flux for real and synthetic refinery wastewater as a function of time.

**Table 1 membranes-12-00161-t001:** The main parameters of the synthetic and real refinery wastewaters.

Parameter	Unit	Value	
		Real	Synthetic
Oil and grease	mg/L	170	200
Chemical oxygen demand (COD)	mg/L	555	398
Total suspended solids (TSS)	mg/L	213	543
Ammonia nitrogen (NH3-N)	mg/L	29.10	27.60
pH	-	6.70	6.70
Temperature	°C	27–28	26–28

**Table 2 membranes-12-00161-t002:** The experiment conditions of refinery wastewater ultrafiltration.

Code	Variables	
	MLSS (mg/L)	Air Bubbles Flow Rate (mL/min)
1	3	1.2
2	3	2.4
3	3	3.0
4	4.5	1.2
5	4.5	2.4
6	4.5	3.0

**Table 3 membranes-12-00161-t003:** Membrane properties and operating characteristics of the submerged ultrafiltration.

Parameter	Membrane
Membrane configuration	Hollow fiber
Membrane material Hydrophilic additive added	PVDF LiCl, TiO_2_
Outer diameter (mm)	1.10
Inner diameter (mm)	0.55
Pore size (nm)	34.05
Porosity	88.42%
Contact angle (^o^)	47
Zeta potential (mV at pH 6.9)	62.00
Tensile strength (MPa)	3.37 ± 0.13
Young’s modulus (GPa)	3.81 ± 0.21
Pure water flux (L m^−2^ h^−1^)	82.95 at 250 mmHg
pH feed solution (pH)	6.70
Air bubbles flow rate (mL/min)	1.20, 2.40, 3.00
Mixed liquor suspended solids concentration (g/L)	3.00, 4.50

**Table 4 membranes-12-00161-t004:** Critical flux test conducted with MLSS of 3 g/L and ABFR of 2.4 mL/min.

Flux (L/m^2^h)	*P_ave_* (mbar)	Lp (L/m^2^h bar)	Δ*P*_0_ (mbar)	*dP/dt* (bar/min)	*R_f_/R_t_* (%)
0	0				0
8.5	3	2.83	3	0.1	23.89
12.9	15	0.86	12	0.2	26.21
25.4	39	0.65	24	0.3	28.02
41.2	64	0.64	25	0.21	28.01
68.8	102	0.67	38	0.25	27.98
99.5	165	0.60	63	0.35	32.96
123.1	250	0.49	85	0.40	35.20
133.1	310	0.43	60	0.25	30.11
139.1	370	0.38	60	0.22	31.12
144.4	507	0.28	137	0.46	39.30

**Table 5 membranes-12-00161-t005:** Membrane properties and operating characteristics of the submerged ultrafiltration.

Parameter of Permeate Removal (%)	Refinery Produced Wastewater	
	Synthetic	Real
Oil and grease	97.35	97.82
COD	88.05	90.30
TSS	99.60	99.81
NH3-N	90.65	91.10

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
