# Peer review of "Critical Flux and Fouling Analysis of PVDF-Mixed Matrix Membranes for Reclamation of Refinery-Produced Wastewater: Effect of Mixed Liquor Suspended Solids Concentration and Aeration"

_membranes, 2022, doi:10.3390/membranes12020161_

Round 1

Reviewer 1 Report

The manuscript on the critical flux and fouling of PVDF mixed matrix membrane could be very important for the readers of Membranes. Unfortunately, in the submitted form it is not acceptable. The manuscript must be rearranged and substantially improved from the both technical and merit base points of view. The graphics are of very low quality (Figs. 2, 3, 9, 11, 12, 14-16). The introduction must be extended with regard to the critical flux and fouling, the text from section 2 should be moved to the introduction. There is no information regarding the vacuum conditions. Membrane preparation must be described in this manuscript as well (even in brief). There is many places in the text with the information "Error! Reference source not found". Why the accuracy of numbers are different (e.g. Tab. 1 - pH values are 6.7 and 6.70)? Units should be SI ones (e.g. Tab. 3 - pure water flux). Numbering of equations is wrong, e.g. line 245 and 247. The references format is mixed (line 266-267). Fig. 4 should be redrawn to get much better quality. Eq. 17 (line 338) is unclear. 
The discussion of results is somehow chaotic and it was difficult to follow all the explanations.

Final recommendation - reconsider after the major revisions.

Author Response

Dear Reviewer,

I have revised the manuscript as the reviewer advised, and have been provided the point-by-point responses from reviewers' comments. Thank you.

Reviewer 2 Report

  1. Add a graphical abstract for the manuscript.
  2. Abstract: Please add quantitative results and provide appropriate reasoning. It is lengthy and reduce the abstract size.
  3. In introduction section, focus of the work is not clear, explain well and needs more references.
  4. [Experimental section] The remarks information for chemicals and testing equipments in this work should be given, for example ( Model, Name of supplier, City, Country).
  5. Check the equation numbered in experimental section ( repeated numbered / wrongly numbered). Revise it.
  6. Figure 3 is not clear, revise it.
  7. There are no error bars on the achieved results. Revise it.
  8. What is the significant outcome for this study?
  9. Found few spell mistakes in the current version. Avoid the typo errors and unnecessary space in the written manuscript

Author Response

Dear Reviewer

I have provided the point-by-point response to reviewers' comments and rewritten the manuscript. Sincerely yours.

Round 2

Reviewer 1 Report

The text was improved to some extend, but still there is a lot of manuscript's sections which must be changed. There are system comments "Error! Reference source not found" and all these expressions must be corrected. This is a role of Authors to prepare the error free text. Moreover, axes of the figures - units are missing, e.g. Fig.2. Fig. 10 - the decimal separator is "." not ",". Figs. 11-17 - they must be redrawn as the quality is below acceptable standards for Membranes.

Author Response

Dear Reviewer,

Thank you for your response that can develop a better manuscript to fulfill the Membranes standard. I hope this revised manuscript could publish in this journal.

Kindy regard

Round 3

Reviewer 1 Report

Still Fig. 11 must be improved.

Author Response

Dear Reviewer,

Firstly, I thank you for your third response and suggestion of my manuscript. I have redrawn figures 11 a and b as suggested. Figures 11 a and b were added with standard error for each code (1-6). I have saved the file of the figure in a special file and will be attached in this link journal.

Thank you and best regards. erna
